# Influence of the Integration of Geopolymer Wastes on the Characteristics of Binding Matrices Subjected to the Action of Temperature and Acid Environments

**DOI:** 10.3390/polym14050917

**Published:** 2022-02-25

**Authors:** Rabii Hattaf, Abdelilah Aboulayt, Nouha Lahlou, Mohamed Ouazzani Touhami, Moussa Gomina, Azzeddine Samdi, Redouane Moussa

**Affiliations:** 1Laboratory of Physics and Chemistry of Inorganic Materials, Faculty of Sciences Aïn Chock, University Hassan II Casablanca, Casablanca 53306, Morocco; azdn.samdi@gmail.com (A.S.); redmoussa@yahoo.fr (R.M.); 2National School of Architecture of Tetouan, Avenue Baghdad, Touabel Soufla, Tetouan 93040, Morocco; aboulayt.abdelilah@gmail.com; 3Laboratory of Mechanics, Faculty of Sciences Aïn Chock, University Hassan II Casablanca, Casablanca 53306, Morocco; nouhalahlou10@gmail.com (N.L.); touazzani2014@gmail.com (M.O.T.); 4CRISMAT UMR6508 CNRS, ENSICAEN, 6 Boulevard Maréchal Juin, CEDEX 4, 14050 Caen, France; moussa.gomina@ensicaen.fr

**Keywords:** fly ash, metakaolin, geopolymer wastes, recycling, fire resistance, acid attack

## Abstract

Recycling geopolymer waste, by reusing it as a raw material for manufacturing new geopolymer binding matrices, is an interesting asset that can add to the many technical, technological and environmental advantages of this family of materials in the construction field. This can promote them as promising alternatives to traditional materials, such as Portland cements, which are not so environmentally friendly. Recent studies have shown that the partial replacement of reactive aluminosilicates (metakaolin and fly ash) up to a mass rate of 50% by geopolymer waste does not significantly affect the compressive strength of the new product. In line with these findings, this paper investigates the effects of aggressive environments, i.e., high temperatures (up to 1000 °C) and acid attacks (pH = 2), on the characteristics of these new matrices. Different techniques were used to understand these evolutions: mineralogical analysis by X-ray diffraction (XRD), thermogravimetry-differential thermal analysis (TGA-DTA), mechanical characterization and scanning electron Microscopy (SEM) observations. The results are very satisfactory: in the exposure temperature range explored, the new matrices containing geopolymer waste suffered losses in compressive strength similar to those of the matrices without waste (considered as materials reference). On the other hand, the new matrices exhibited good chemical stability in acid media. These results confirm that the reuse of geopolymer waste is a promising recycling solution in the construction sector.

## 1. Introduction

In the building and civil engineering sectors, geopolymers are considered a promising alternative to conventional materials based on Portland cements [1,2]. These materials hold the attention of scientists and engineers owing to their numerous technical, technological and environmental advantages. These environmentally friendly materials are characterized by a simple implementation protocol at temperatures below 100 °C (meaning a low carbon footprint), great diversity and availability of raw materials, good properties in the fresh state, excellent mechanical resistance in the hardened state, high fire resistance, good resistance to acids and sulphates [3,4,5,6,7,8] and excellent immobilization capability of toxic and radioactive elements [9,10,11].

These inorganic materials with amorphous behavior under X-ray have a microstructure made up of a three-dimensional stack of aluminate (AlO_4_) and silicate (SiO_4_) tetrahedra, with the insertion of alkaline or alkaline-earth cations (generally Na^+^, K^+^ and Ca^2+^) compensating charges. They are generally synthesized by the reaction of a solid reactive aluminosilicate precursor (fly ash, metakaolin etc.) with an alkaline solution of sodium or potassium silicate at a temperature below 100 °C [1,12,13]. 

Several authors agree on the fact that this synthesis process generally involves three essential steps: (a)—dissolution of the starting reactive aluminosilicate precursors under the effect of the alkaline solution to form aluminate and silicate monomers, (b)—oligomerization and formation of aluminosilicate gel, and (c)—polycondensation and precipitation of the geopolymer phase [14,15,16]. The physicochemical characteristics and the performance of the final materials strongly depend on parameters, such as the nature and the composition of the starting reactive aluminosilicates [17,18,19], the composition and the concentration of the alkaline activation solution and the curing regime (temperature, duration and humidity) [14,20,21,22].

The gradual development of these materials in the building and civil engineering sectors will result in the generation of increasing amounts of waste (manufacturing scrap and end-of-life materials). Recycling is one of the best management strategies to give value to this type of non-biodegradable waste in order to avoid its accumulation in landfills. This solution is in line with the spirit of several contemporary sustainable development policies concerning the recovery and reuse of waste at the end of its life cycle, in particular the European Union action plan on the circular economy: “less waste, more value” [23].

Few works dedicated to this theme have been identified in the literature [24,25,26,27,28,29]. In a recent study [24], we investigated the possibility of partially replacing fly-ash (FA) or metakaolin (MK) with Fly-ash-based Recycled Geopolymer (FARG) and Metakaolin-based Recycled Geopolymer (MKRG) wastes in order to manufacture new geopolymer matrices that consume less raw materials. Fine-grained powders of FARG and MKRG were used in the partial substitution of the sources of aluminosilicates FA and MK up to a mass rate of 50% to produce three formulations of binders referenced as FA/FARG, MK/MKRG and MK/FARG. 

Physicochemical and mechanical characterization revealed they are highly homogeneous with good behavior in the fresh state (flowability), good coating ability of mineral aggregates and high mechanical performances in the hardened state comparable to those of materials without waste, even at the highest replacement rates. These performances qualify them to meet the requirements of the standards of many building materials.

This study is part of the work to improve the reliability of geopolymer materials that we have been undertaking for some time. The objective here is to further characterize the new matrices by studying their durability, in terms of their behavior with respect to aggressive environmental agents (temperature and acid solutions).

Several comparative studies have shown that geopolymer binders have better fire resistance than Ordinary Portland Cement (OPC) [7,11,30,31,32,33,34]. In fact, cement is a hydraulic binder: part of the mixing water reacts chemically to form hydrated phases (hydrated calcium silicate C-S-H, calcium hydroxide Ca(OH)_2_ etc.). These phases dehydrate between 400 and 550 °C, which can cause the formation of pores and the creation of cracks. This deep microstructural damage leads to a drastic degradation of the mechanical performance. In contrast, the microstructure of geopolymers is more stable because it only undergoes decompositions at higher temperatures.

The thermal performances of geopolymers are strongly governed by the chemical and mineralogical nature of the aluminosilicate precursors used for their production [30,35,36,37,38,39,40,41]. Kong et al. [39] reported that fly-ash-based geopolymers exhibit better thermal performance than those based on metakaolin after exposure to 800 °C. This is due to the high content of interconnected micropores in the fly-ash-based matrices, whose presence facilitates the drainage of water during heating. In addition, the onset of sintering of unreacted particles increases the mechanical strength. In contrast, the metakaolin-based matrix has isolated and larger pores, which makes it more difficult for water to escape and causes greater damage.

The properties and composition of the activating solution also have a significant impact on the thermal performance of geopolymer materials. Thus, the nature of the microstructure of materials is determined by the amount of soluble silica provided by the solution and influences their performance, both before and after exposure to high temperatures [40,42,43,44]. Lahoti et al. [44] studied the thermal behavior of geopolymers based on F-type fly-ash activated by sodium silicate solutions of different compositions. 

The residual compressive strengths of materials exposed to 900 °C show drastic variations. On the one hand, an 80% reduction was associated with the formation of cracks due to the combined effects of the evaporation of moisture water and of the crystallization of nepheline, a particular sodium aluminosilicate. On the other hand, a 150% improvement was attributed to the densification of the matrix by the formation of inter-particle bonds due to viscous sintering.

The effects of the nature and the concentration of the alkali cation on the thermal performance of geopolymers have been reported in several works [31,45,46,47]. The geopolymers produced by activators based on potassium silicate show better thermal stability than those obtained from sodium silicate [45]. Moreover, an excessive content of alkali cations lowers the recrystallization temperature of certain phases, such as nepheline, which affects the mechanical performance of the materials [1,44,48].

Another important property that characterizes these materials is their good resistance to acidic media. Several works have focused on the study of this phenomenon [11,49,50,51,52,53,54]. Although the experimental conditions varied from one study to another, in terms of the concentration of the etching solution used, the nature of the acids (HCl, H_2_SO_4_ or acetic acid), the results showed that geopolymers are generally chemically stable with relatively low short-term mechanical strength losses. 

The main mechanism causing loss of resistance in an acidic environment is identified with the replacement of the alkaline cations (Na^+^ or K^+^) in excess in the material by H^+^ protons. Then the de-alumination of the geopolymer structure follows. This leads to an increase in the pH of the etching batch and to the formation of a lacunar siliceous structure that is less mechanically resistant [50,53]. 

This work is divided into two parts. The first is devoted to the study of the behavior of new binders after exposure to 200, 400, 600 and 800 °C for 2 h. This behavior is evaluated by DTA–TGA analysis. The second part is dedicated to the evaluation of the characteristics of these new binders subjected for 28 days to the chemical action of acid media. After these various treatments, the residual compressive strengths were measured, microstructural analyses by X-ray diffraction and observations by optical microscopy and by scanning electron microscopy were also performed. 

## 2. Materials and Methods

### 2.1. Raw Materials and Constituents

The starting materials were:-Aluminosilicate powders: a commercial metakaolin (MK, from Imerys-France) and fly-ash from coal combustion (FA, from the Jarf Lasfar thermal power station in Morocco).-An alkaline activating solution (AAS) with SiO_2_/Na_2_O molar ratio = 1.2 and 63% by weight of water. This solution is prepared from a mixture of sodium hydroxide (98% purity), commercial sodium silicate (SiO_2_/Na_2_O molar ratio of 2) and water.-Hydrochloric acid (37% HCl) was mixed with distilled water to prepare the acid solutions.

The chemical composition of the starting reactive aluminosilicates MK and FA indicate that they consist mainly of silica SiO_2_ and alumina Al_2_O_3_, with small amounts of Fe_2_O_3_ and CaO in FA (Table 1). The amount of SO_3_ comes from the sulphates present in the material (CaSO_4_, in particular). The mineralogical analysis of MK and FA by DRX (Figure 1) highlights the presence of an intense dome between 15 and 40°, which indicates the existence of an amorphous aluminosilicate phase, with the presence of quartz and residual kaolinite in MK and mullite and quartz in FA. The D_90_ particle size distribution parameters are 24.64 and 80.2 μm for metakaolin and fly ash, respectively (Figure 2).

### 2.2. Material Manufacturing

The manufacture of the materials consists of three stages [24]: The first stage is preparing the reference geopolymer binders. To this end, two types of materials were developed:-A metakaolin-based geopolymer, MKref, obtained by mixing metakaolin powder (MK) and alkaline activating solution (AAS) at a Liquid/Solid ratio (L/S) ratio of 0.83. This mixture is hardened at 60 °C for 5 h.-A fly-ash-based geopolymer, FAref, obtained by mixing fly-ash powder (FA) and alkaline activation solution (AAS) at an L/S ratio of 0.58. This mixture is cured at 80 °C for 20 h.To simulate geopolymer waste, the manufactured materials (MKref and FAref) are stored for one year in the open air at room temperature, then they are ground to obtain a fine powder D < 80 µm (Figure 3) referenced MKRG for materials based on metakaolin and FARG for those based on fly-ash.The third stage consists in preparing the new matrices by substituting the starting aluminosilicates (fly-ash and metakoalin) with the waste (MKRG and FARG) up to a mass rate of 50%. These mixtures are then subjected to the action of the activation solution under the conditions described in stage 1. Three systems were, thus, prepared: FA/FARG, MK/MKRG and MK/FARG. Note that the materials referenced MKref and FAref are fresh matrices with 0% waste. Table 2 summarizes the data for the manufacturing of the materials.

Analyses of the diffractograms (Figure 4) of MKref and FAref and those of the geopolymer waste (FARG and MKRG) do not indicate any change in the mineralogical composition. We note the presence of a large vitreous dome between 15 and 40° that testifies to the presence of a large quantity of amorphous phase that characterizes the geopolymeric phase, as well as that of crystallized phases coming from the starting aluminosilicates and resisted alkaline attack. These are quartz and mullite for fly-ash-based materials (FAref and FARG) and residual quartz and kaolinite for metakaolin-based materials (MKref and MKRG) [22,24,25].

### 2.3. Fire Resistance

The fire resistance of geopolymer matrices was evaluated according to a protocol used in the literature [31,55]. The samples were first stored for 28 days after demolding. A control batch was stored at room temperature in air. A second batch of samples was dried at 105 ± 1 °C for 24 h in an electric muffle oven, and then it was heated up to the treatment temperature (at a rate of 6.67 °C/min) for a dwell of 2 h before naturally cooling down to room temperature inside the oven in order to avoid thermal shock. Four treatment temperatures were selected: 400, 600, 800 and 1000 °C (Figure 5). These specimens were loaded in compression to determine the compressive strengths.

### 2.4. Resistane to Acid Attack

The behavior of these matrices in an acid environment was previously investigated [50]. After demolding, the samples were dried for 24 h at room temperature in air and then immersed for 28 days in various solutions maintained at 25 °C in hermetic plastic bags (Figure 5). The ratio of the sample mass on the volume of the solution was 0.1 g/mL. The solutions were:-HCl solution (AD) with pH = 2. This solution was renewed weekly for 4 weeks.-HCl solution (AC) with pH = 2. This solution was not renewed during the treatment period.

### 2.5. Characterization Methods

-The basic chemical composition of the powders was determined by X-ray fluorescence by using a Thermo ARL 9800XP spectrometer equipped with an X-ray tube with 3 kW maximum power (30 kV and 80 mA).-The mineralogical composition of the materials was determined using XRD on powder, using a Bruker D8 copper anticathode diffractometer (CuKα 2θ (λKα = 1.5418 Å) operating at 40 mA and 40 kV. The results were collected for 2θ in the range 10 to 70° with an increment of 0.059°. The identification of the crystalline phases was made using version 2.1 of the DIFFRAC EVA software.-Differential thermal analysis (DTA) and thermogravimetric analysis (TGA) were performed by using a Shimadzu DTG-60 H instrument operating at a heating rate of 6.67 °C/min.-The fracture surfaces of the materials before and after the chemical treatment were analyzed using a Tescan VEGA 3 scanning electron microscope and Keyence ZHX-700 type digital optical microscope.-Compressive strength testing was conducted on a Perrier type tensile machine, equipped with a 200 kN capacity loading cell. Four tests were performed for each material formulation.-The pH of the AC and AD acid media was determined using a standard pH-measurement device. Three pH measurements were performed for each material formulation in each medium.

## 3. Results

### 3.1. Fire Resistance

#### DTA-TGA

TGA-DTA thermal analyses are suitable techniques for determining the various phenomena that occur in a material subjected to the action of heat. Analyses were performed between room temperature and 1000 °C on samples from the three systems FA/FARG, MK/MKRG and MK/FARG (Figure 6). 

DTA/TGA analyses show that FARG and MKRG materials simulating waste exhibit the same thermal behavior as MKref and FAref reference materials in the fresh state with the exception of water loss slightly higher for the fresh materials. Regardless of the nature of the sample, most of the measured mass loss on the TGA curves occurs below 300 °C. This loss is marked on the DTA curves by a strong endothermic effect around 75 °C, which is assigned to the loss of free water [30,41]. In the particular case of the metakaolin-based materials, MK/MKRG and MK/FARG (Figure 6b,c), this effect includes a second endothermic peak with lower intensity around 125 °C, which is attributed to the dehydration of zeolite structures present in the geopolymer gel [56,57]. 

Beyond 300 °C, two distinct behaviors are observed: the samples of the MK/MKRG system do not undergo any mass loss (Figure 6b), whereas those of the FA/FARG or MK/FARG system (Figure 6a,c) show a second loss, weaker and marked by an endothermic effect centered around 400 °C, which is attributed to the dehydroxylation of portlandite Ca(OH)_2_ [41,54]. Finally, we note that the increasing addition of recycled geopolymers, to the detriment of precursor sources of aluminosilicate, increases the overall mass loss. This could be explained by the effect of substituting FA and MK materials with others (FARG and MKRG) with a higher loss on ignition (Table 1).

### 3.2. Mineralogical Analysis

XRD mineralogical analysis has been used to demonstrate phase transformations that occur after exposure to different temperatures. The results recorded before and after heat treatment at 400, 600, 800 and 1000 °C are shown in Figure 7.

It is noted that the diffractograms of the reference matrices FAref and MKref (without waste), as well as those of the samples with 50 wt.% of waste, presented, before heat treatment, a large glass dome in the range 17 < 2θ < 33°, which characterizes the amorphous geopolymer phase [20,24]. Analysis also reveals the presence of crystalline phases, such as quartz and mullite in fly-ash-based matrices (Figure 7a,b), and quartz and residual kaolinite in metakaolin-based matrices (Figure 7c–e). No microstructural change was observed after exposure of these materials to 400 °C, which reflects the good thermal stability of the geopolymer gel at this temperature.

Moreover, the treatment at 600 °C has different consequences depending on the nature of the matrices. We notice the crystallization of a new phase, nepheline NaAlSi_3_O_8_ in fly-ash-based materials, the characteristic peaks of which are amplified following the introduction of waste. This phenomenon is not observed in binders based on metakaolin. This is attributed to the difference in the chemical composition of the starting aluminosilicates: unlike metakaolin, fly-ash contains appreciable amounts of the oxides Fe_2_O_3_ (5.80%) and K_2_O + Na_2_O (2.66%) (Table 1). These oxides are known for their role as fluxes and mineralizing agents that contribute to the crystallization of nepheline [1,40,48,58]. The substitution of fly-ash by FARG increases the contents of these oxides in the materials of the FA/FARG system, which favors the neo-crystallization phenomenon in these matrices.

At the highest temperatures, all the exposed materials are subjected to important transformations: amplification of nepheline crystallization at the expense of the amorphous geopolymer phase (disappearance of the glassy dome) and gradual dissolution of quartz and mullite. At 1000 °C, the materials appear to be exclusively composed of nepheline.

### 3.3. Mechanical Behavior

The mechanical behavior of the matrices was studied by analyzing their compressive strengths in order to demonstrate the effect of the addition of recycled geopolymers on the performance of the materials exposed to temperatures of 400, 600, 800 and 1000 °C. The results are shown in Figure 8. To better assess the effect of the two factors, namely the substitution rate and the temperature, the evolution of the normalized residual compressive strength is depicted in Figure 9.

Prior to the heat treatment implementation, the MKref and FAref reference matrices had compressive strengths of 63 and 54 MPa, respectively. The high content of reactive amorphous phases of the starting aluminosilicates (fly-ash and metakaolin) and the adequate activation conditions applied having favored the phenomenon of geopolymerization, a compact microstructure was obtained (as can be observed on the fractographies shown in Figure 10) as well as high compressive strengths [14,20]. 

The introduction of recycled geopolymer waste up to a substitution rate of 40 wt.% at the expense of reactive aluminosilicates had no noticeable effect on the strength of the new matrices. This behavior is due to the establishment of interfacial bonds between the gel of the newly formed geopolymer and the particles of integrated waste. 

For a substitution rate of 50 wt.%, a reduction of about 20% in the compressive strength occurs although the observations by optical microscopy and SEM of the fracture surfaces of the specimens show a compact structure similar to that of the reference matrices (Figure 10 and Figure 11). These features are assigned to the reduction in the volume of the neo-formed geopolymer gel, which is the source of the material’s cohesion. A phenomenological model was previously proposed to explain this good behavior and the mechanisms governing the good adhesion between the reference matrices and the geopolymer waste aggregates [24].

The trend of compressive strength variation as a function of the treatment temperature is in line with the data reported by A. Celik [59] and M. Lahoti [43] for metakaolin-based geopolymer matrices and by X. Jiang [41] for fly-ash-based materials. The increase in the temperature of the heat treatment results in a degradation of the mechanical strength of the matrices. However, there is a weak influence of increasing the exposure temperature for additions less than 40 wt.% (Figure 8 and Figure 9).

Exposure to 400 °C causes a significant decrease in mechanical strength of about 40%, regardless of the system studied. This reduction is mainly due to the increase in porosity caused by the escape of free and zeolitic water revealed by TGA/DTA (Figure 6) and confirmed by microscopic and naked eye observations that reveal the development of fine microcracks within the matrices (Figure 10 and Figure 11) [7,41,44]. 

The treatment at 600 °C causes approximately 50% degradation of the mechanical strength of the materials based on metakaolin (MK/MKRG and MK/FARG), which is lower than that (of about 60%) recorded for the fly-ash-based materials (Figure 8 and Figure 9). This difference is attributed to the decomposition of the geopolymer binding phase due to the early crystallization of nepheline that occurs in the fly-ash-based matrices (Figure 7) [7,32].

Exposure to 800 or 1000 °C further decreases the mechanical strength due to the significant degradation of the microstructure of the various matrices. Indeed, a pronounced development of the porous phase and the appearance of macrocracks (Figure 10 and Figure 11) resulting from the total transformation of the binding geopolymer phase into nepheline, are noted (Figure 7) [7,43,44].

### 3.4. Resistane to Acid Attack

Figure 12 shows the evolution of the pH of the material immersion baths over four weeks, for the two acidic mediums AC and AD. In the case of AC solutions, the evolution of the pH appears to be independent of the nature of the immersed material: the pH of the solutions increases rapidly and reaches the base range during the first week before stabilizing beyond the third week. It is also noted that the pH values increase when the waste rate in the materials is increased. This development indicates a strong and rapid interaction of the material with the acid bath. 

According to J. Kwasny et al. [53], the increase in pH is due to the substitution of the Na^+^ and K^+^ alkaline cations of the geopolymers by the H^+^ protons of the acidic medium. This mechanism is discussed by X.X. Gao et al. [50]. According to these authors, on contact with the acid solution, the materials release alkaline species present in excess after geopolymerization. 

These species then form compounds of a basic nature. This is because the excess alkali hydroxide in materials reacts with carbon dioxide present in the air to form carbonate species of the alkaline element, such as NaHCO_3_ and Na_2_CO_3_, that dissolve in solution and neutralize the H^+^ proton. The increase in the pH of solutions after immersion of materials containing waste can therefore be explained by the gradual increase in the amount of alkaline species provided by the waste. 

Analysis of the pH in the AD medium provides further lessons. Indeed, all the baths analyzed from the third week had an acidic pH, which indicates that the materials release less basic species in the solutions. This suggests that the neutralization of basic species by the acidic solution is limited in time [50].

Figure 13 shows the X-ray diffractograms of the geopolymeric materials of the three systems before and after acid attack for four weeks. No change from non-immersed materials (treated at room temperature) was noted. These results are explained because the amorphous phase of the geopolymers and the crystalline phases were not affected and no new phase was detected. They are in good agreement with other works that emphasized the chemical stability of cross-linked structures of geopolymeric binding matrices [50,51,52,53,54] in aggressive acidic environments. They also prove that the substitution of reactive aluminosilicates by recycled geopolymers had no significant effect on the chemical stability of the newly formed matrices.

In order to better assess the impact of the acid attack after four weeks immersion, the changes in compressive strength and normalized strength were determined (Figure 14 and Figure 15).

It is noted that regardless of the system and the substitution rate considered, the immersion of materials in acid solutions led to about a 20% reduction in strength compared to non-immersed materials. These losses in mechanical strength are similar to those noted for the reference materials (without waste) and of the same order as those reported in the literature [11,50,51,52,53,54]. They are attributed to a de-alumination mechanism of the geopolymer phase, which results in the formation of a lacunar siliceous structure [50,53]. However, this phenomenon appears to be very limited compared to the low mass losses noted after the acid attack (1.4% at most). This observation is confirmed by the optical microscopy analyses of the fracture surfaces (Figure 16), which show that the matrices after immersion are similar to those not attacked (a small increase in the size of the pores).

## 4. Conclusions

This work is a continuation of the studies we are conducting on the recyclability/recycling of geopolymer materials. The present study was performed to assess the impact of the integration of geopolymer waste up to 50 wt.% on fire resistance, and resistance toward chemical attack by immersion in acidic environments was also assessed. The results indicate that:-Regardless of the substitution rate, the hardened materials showed, at room temperature (RT), good mechanical performance with compressive strengths greater than 45 MPa.-Regardless of the waste substitution rate, the materials showed good fire resistance similar to that of waste-free materials. They maintained compressive strength greater than 20 MPa after treatment at 600 °C. The degradations observed included increased porosity resulting from the release of residual water and the crystallization of nepheline, which depletes the binding phase.-The crystallization of nepheline was earlier (600 °C) in the fly-ash-based materials FA/FARG. This is due to the high content of Fe_2_O_3_, K_2_O and Na_2_O oxides in the starting fly-ash, known for their role as fluxes and mineralizing agents. However, this phenomenon was only observed from 800 °C for the metakaolin-based materials MK/MKRG and MK/FARG.-Regardless of the waste substitution rate, the materials demonstrated very good resistance to acid attack (pH = 2) with resistance losses not exceeding 28%.-The increase in the pH of the attack baths confirms the mechanism of release of sodium ions present in excess in the materials, which enriches the baths in alkaline species by their combination with atmospheric CO_2_.-The materials were chemically stable in the media studied. Indeed, no significant impact on the mineralogical composition nor on the mass losses was recorded.-The increase in the rate of substitution of geopolymer waste had the effect of increasing the concentration of excess sodium ions in the material and, thus, that of the pH.

These results are encouraging and motivating for safeguarding the environment through the reuse of geopolymer wastes as additives in geopolymeric building materials. 

## Figures and Tables

**Figure 1 polymers-14-00917-f001:**
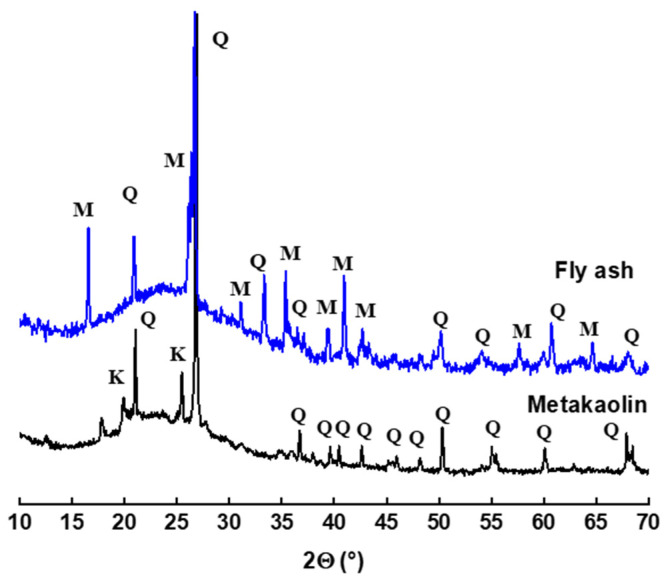
Metakaolin and fly-ash X-ray diffractograms (Q: Quartz; K: Kaolinite; and M: Mullite).

**Figure 2 polymers-14-00917-f002:**
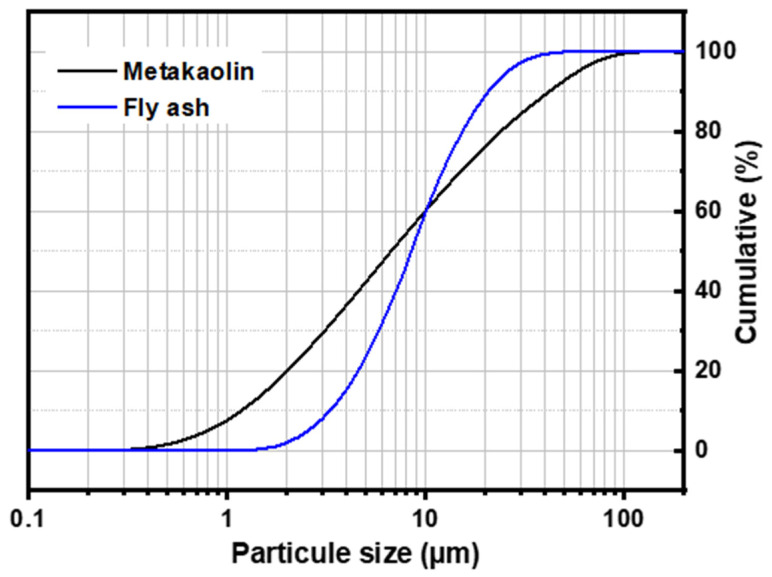
Particle size distribution of metakaolin and fly-ash powders.

**Figure 3 polymers-14-00917-f003:**
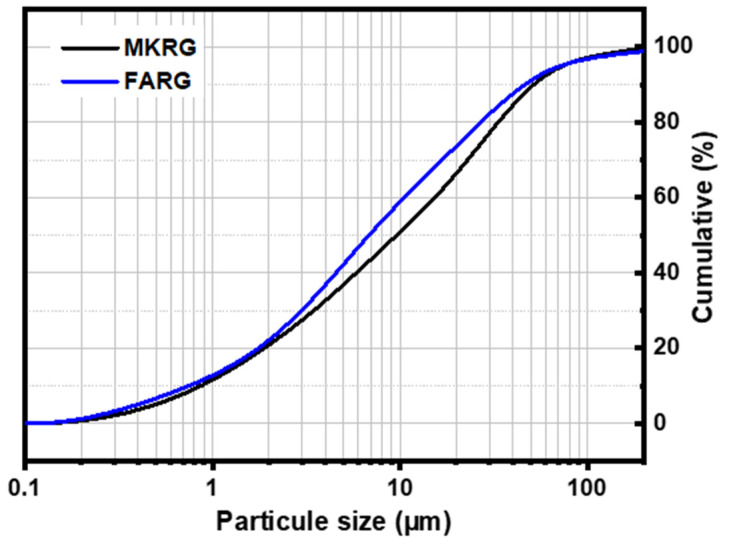
Particle size distribution of MKRG and FARG powders.

**Figure 4 polymers-14-00917-f004:**
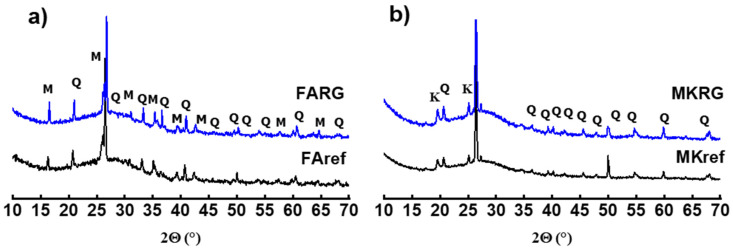
X-ray diffractograms of materials FAref and FARG (**a**) and MKref and MKRG (**b**) (Q: Quartz; K: Kaolinite; and M: Mullite).

**Figure 5 polymers-14-00917-f005:**
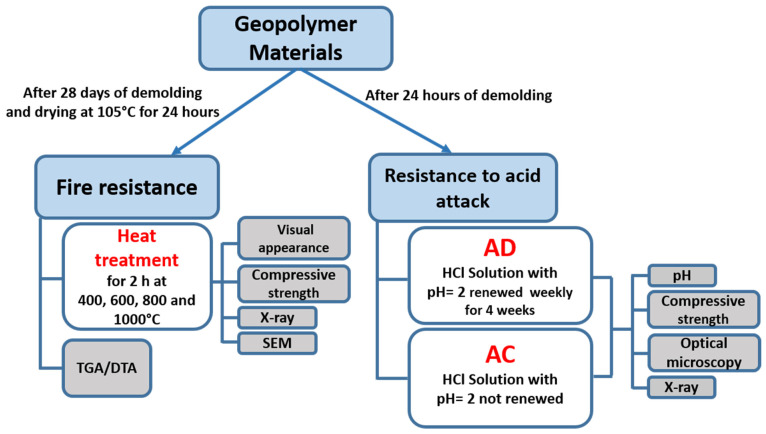
Treatment protocol for geopolymeric materials.

**Figure 6 polymers-14-00917-f006:**
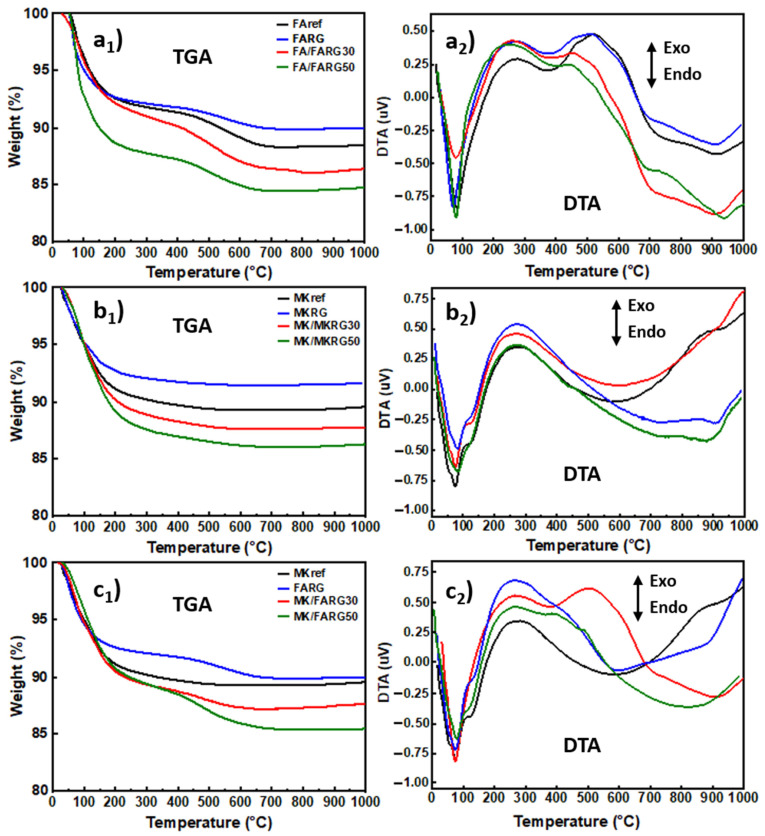
TGA/DTA of the materials as a function of the recycled geopolymer rate for FA/FARG (**a**), MK/MKRG (**b**) and MK/FARG (**c**).

**Figure 7 polymers-14-00917-f007:**
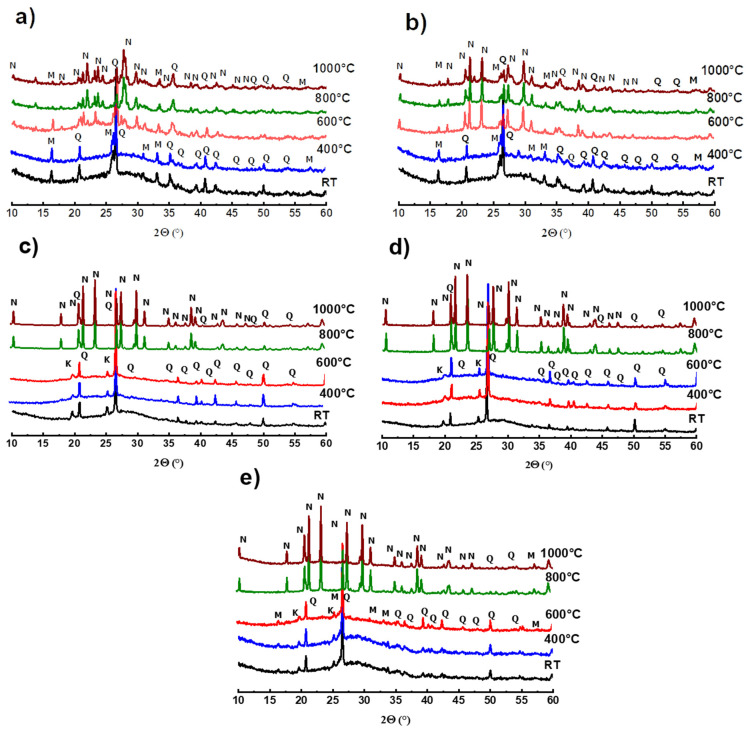
X-ray diffractograms at different temperatures, as a function of the recycled geopolymer rate: FAref (**a**), FA/FARG50, (**b**) MKref (**c**) MK/MKRG50 (**d**) and MK/FARG50 (**e**) (Q: Quartz; K: Kaolinite; M: Mullite; and N: Nepheline).

**Figure 8 polymers-14-00917-f008:**
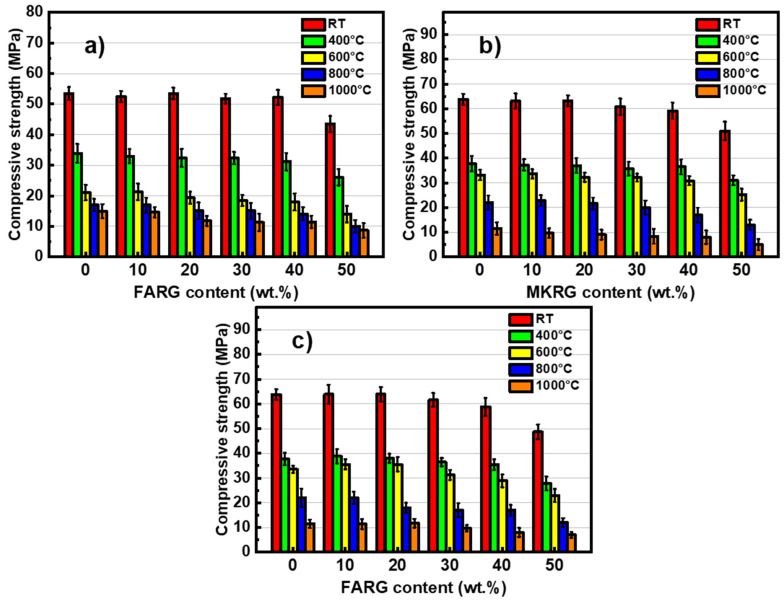
Variation of the compressive resistance as a function of the recycled geopolymer rate at different temperatures: FA/FARG (**a**), MK/MKRG (**b**) and MK/FARG (**c**).

**Figure 9 polymers-14-00917-f009:**
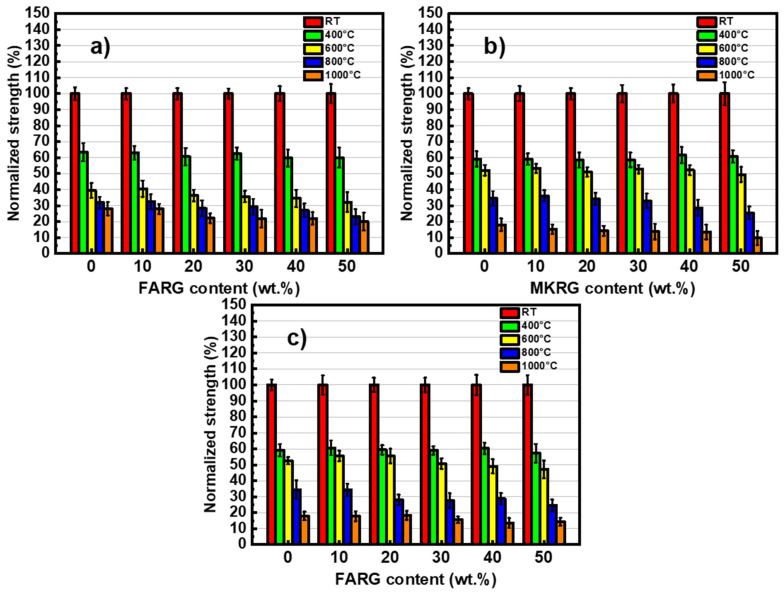
Variation of the normalized compressive strength as a function of the recycled geopolymer rate at different temperatures: FA/FARG (**a**), MK/MKRG (**b**) and MK/FARG (**c**).

**Figure 10 polymers-14-00917-f010:**
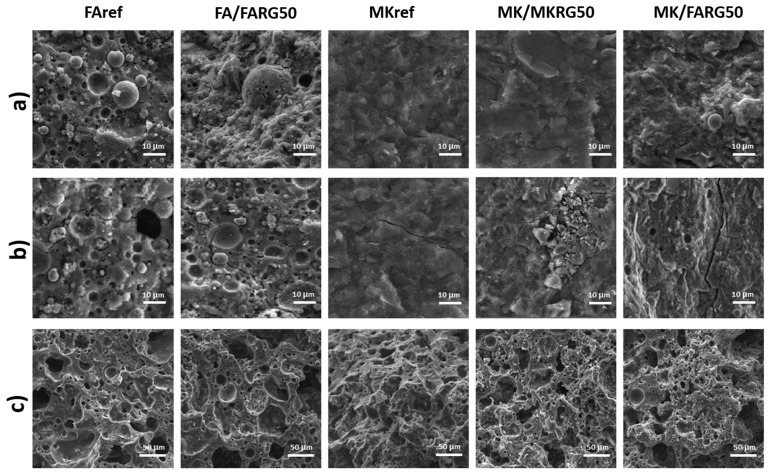
Micrographs of the fracture surfaces as a function of the recycled geopolymer rate at different temperatures: RT (**a**), 400 °C (**b**) and 800 °C (**c**).

**Figure 11 polymers-14-00917-f011:**
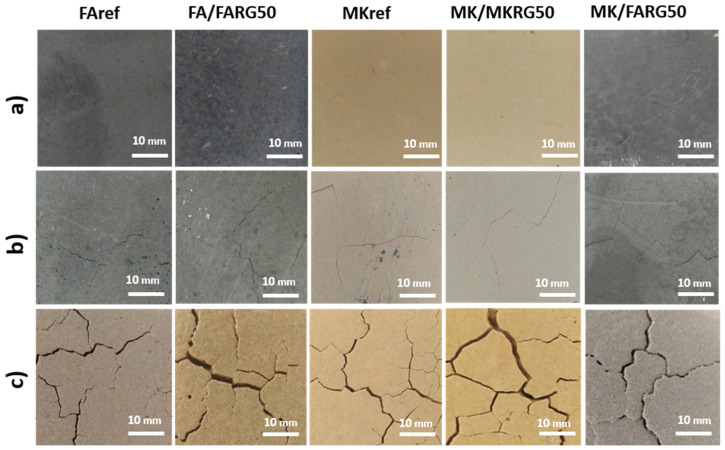
Surfaces of the materials as a function of the recycled geopolymer rate at different temperatures: RT (**a**), 400 °C (**b**) and 800 °C (**c**).

**Figure 12 polymers-14-00917-f012:**
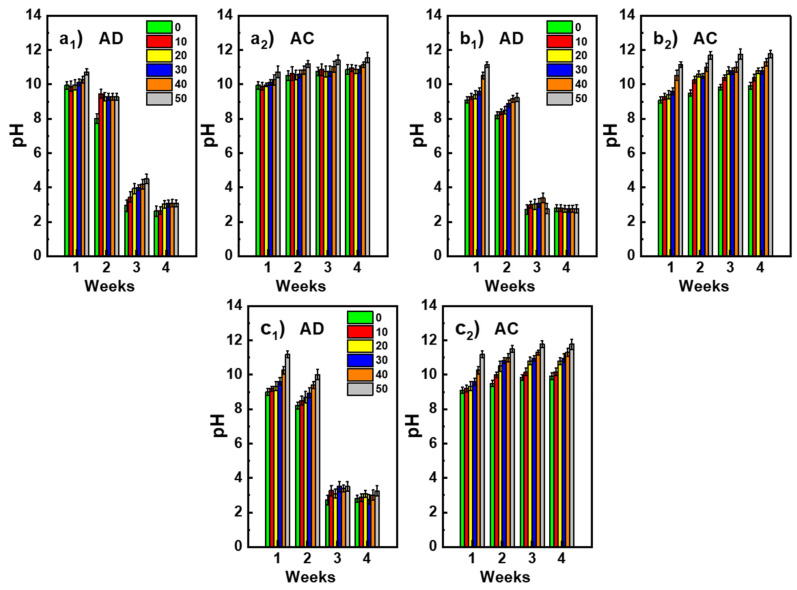
Variation of pH as a function of the rate of recycled geopolymer and the immersion duration in medium AC or AD: FA/FARG (**a**), MK/MKRG (**b**) and MK/FARG (**c**).

**Figure 13 polymers-14-00917-f013:**
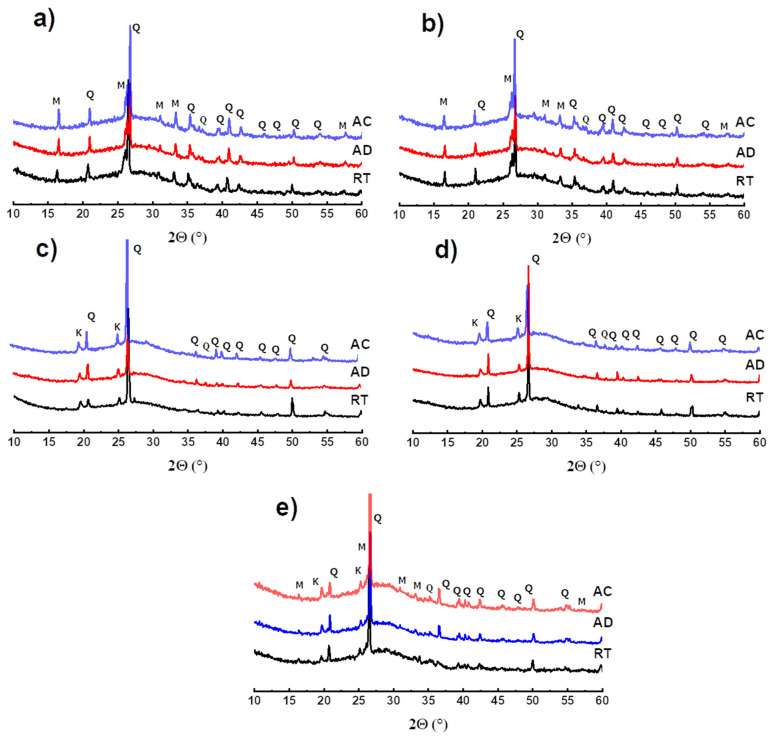
X-ray diffractograms of materials FAref (**a**), FA/FARG50 (**b**), MKref (**c**), MK/MKRG50 (**d**) and MK/FARG50 (**e**) prior to (at RT) and after acid attack (Q: Quartz; K: Kaolinite; and M: Mullite).

**Figure 14 polymers-14-00917-f014:**
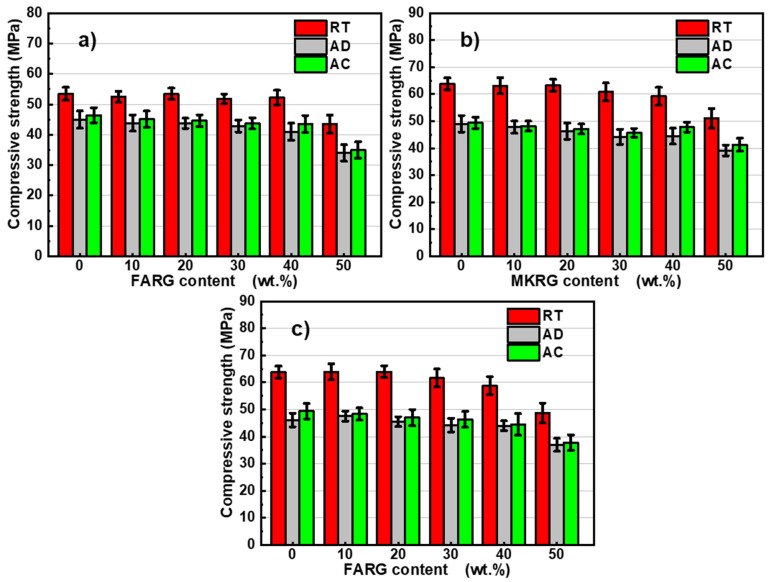
Variation of the compressive strength of the materials in different media as a function of the recycled geopolymer rate: FA/FARG (**a**), MK/MKRG (**b**) and MK/FARG (**c**).

**Figure 15 polymers-14-00917-f015:**
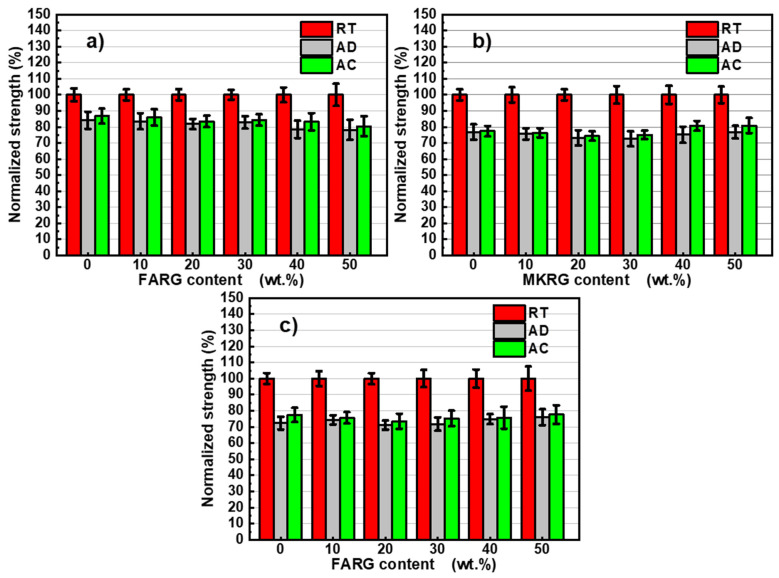
Variation of the normalized compressive strength of the materials in different media as a function of the recycled geopolymer rate: FA/FARG (**a**), MK/MKRG (**b**) and MK/FARG (**c**).

**Figure 16 polymers-14-00917-f016:**
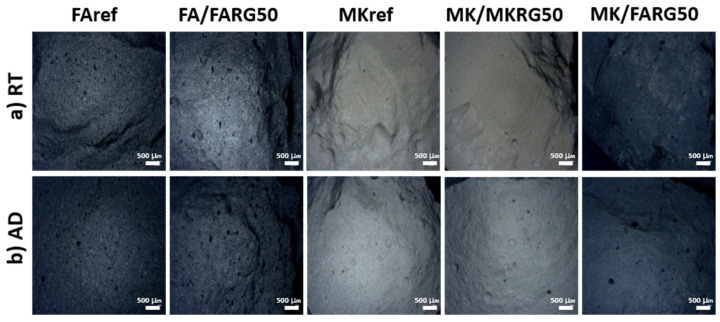
Optical microscope observations of the materials as a function of recycled geopolymer rate: at room temperature (**a**) and in different medium AD (**b**).

**Table 1 polymers-14-00917-t001:** Quantitative chemical composition of the powders (LOI: loss on ignition).

Oxide (wt.%)	SiO_2_	Al_2_O_3_	Fe_2_O_3_	CaO	MgO	SO_3_	Na_2_O	K_2_O	P_2_O_5_	LOI
MK	56.27	36.39	1.38	0.42	0.35	0.19	0.17	0.79	0.06	2.04
MKRG	54.94	25.59	1.10	1.23	0.29	0.31	8.14	0.68	0.10	8.57
FA	50.99	22.36	5.80	4.75	2.19	0.51	0.89	1.77	0.85	9.90
FARG	46.23	17.04	5.39	5.36	2.09	0.55	6.52	1.64	0.57	12.88

**Table 2 polymers-14-00917-t002:** Data used for the manufacturing of the geopolymer materials.

	AAS	Liquid/Solid Ratio (L/S)	Curing Temperature (°C)	Curing Duration (h)
FA/FARG	1.2	0.58	80	20
MK/MKRG	1.2	0.83	60	5
MK/FARG	1.2	0.83	60	5

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
