# Peer review of "Influence of the Integration of Geopolymer Wastes on the Characteristics of Binding Matrices Subjected to the Action of Temperature and Acid Environments"

_polymers, 2022, doi:10.3390/polym14050917_

Round 1
Reviewer 1 Report
- Lines 160-163. In this study, geopolymer waste was a simulated geopolymer waste (MKref and FAref), which was stored for one year in the open air at room temperature. The storage conditions should be specified. For example, for practical application, the temperature fluctuation and storage environment may have significantly affected the characteristics of geopolymer waste. The change of freshly prepared MK/geopolymer and FA/geopolymer characteristics and simulated geopolymer waste should be compared.
- The particle size of geopolymer waste is an important factor in this study. The particle size should be specified in the paper.
- Figure 2 showed the TGA-DTA thermal analyzes for determining the various phenomena that occur in a material subjected to the action of heat. This loss is attributed to the loss of hygroscopic water. The reviewer suggested that the weight loss estimated from the TGA curve can be used to determine the water content of various samples.
- Figure 2. The reviewer suggested that the TGA-DTA thermal analyzes for fresh geopolymer and related composite (freshly prepared MK/geopolymer and FA/geopolymer) should be included.
- Similar comments (comment 4) applied to Figures 3-11.
- Lines 335-337. The authors explained that the increase in the pH of solutions after immersion of materials containing waste can be explained by the gradual increase in the amount of alkaline species provided by the waste. The reviewer suggested that additional experiments can be conducted to measure the content of alkaline ions in the solution to verify the statement.
- Line 355. Figure 8 --> Figure 9.
Author Response
Review #1
Response to note1/ About the storage conditions of fresh geopolymers in order to simulate the production of waste, we naturally stored the materials in a room in the open air and at room temperature.
According to the literature (a, b and c), geopolymer materials stored under similar conditions in terms of temperature and storage time are chemically stable; the only change likely to occur is the appearance of a very small amount of carbonate, cancrinite and franzinite.
In our case, after storage, no new phase was detected by the two analytical techniques used (DRX and DTA/TGA).
We have therefore added in the text (in the materials preparation section) the diffractograms of the MKref and FAref materials (these are the fresh matrices without waste) and those of the geopolymer waste (FARG and MKRG) (Figure 4), accompanied by a comparative description (lines 249-251).
We have also given comments on the DTA/TGA analyzes of FARG and MKRG materials simulating waste and those of MKref and FAref (lines 249-251).
- Sun, Z.; Vollpracht, A. One year geopolymerisation of sodium silicate activated fly ash and metakaolin geopolymers. Cem. Concr. Compos. 2019, 95, 98–110, doi:10.1016/j.cemconcomp.2018.10.014.
- Zhang, Z.; Provis, J.L.; Ma, X.; Reid, A.; Wang, H. Efflorescence and subflorescence induced microstructural and mechanical evolution in fly ash-based geopolymers. Cem. Concr. Compos. 2018, 92, 165–177, doi:10.1016/j.cemconcomp.2018.06.010.
- Longhi, M.A.; Rodríguez, E.D.; Walkley, B.; Zhang, Z.; Kirchheim, A.P. Metakaolin-based geopolymers: Relation between formulation, physicochemical properties and efflorescence formation. Compos. Part B Eng. 2020, 182, 107671, doi:10.1016/j.compositesb.2019.107671.
Response to note 2/ To respond to your suggestion, we have added the powders particle size distribution curve (figures 2 and 3).
Response to note 3/ Indeed. To this end, we have indicated the evolution of the total loss as a function of the substitution rate, in the last point of the “DTA-TGA results” paragraph. In fact, as can be seen on the TGA curves, the loss of water at a temperature below 300°C represents the main part of the overall loss for the three systems.
Response to notes 4/ and 5/ In the preparation section, we have clarified that the reference materials are fresh MKref and FAref materials (matrices without waste). Associated curves are included in the cited figures.
Response to note 6/ We undertook a qualitative approach in order to evaluate the effect of the acid attack on the materials only by the measurement of the immersion baths pH variation. We had estimated that the determination of the amount of alkaline ions (relevant in the case of a quantitative study) on the attack reaction kinetics for example, would not be of a significant contribution in our case. Our approach allowed confirming the following facts:
- The acid attack quickly results in a significant increase in the pH of the baths. This increase confirms the results reported in the literature (Ref. X.X. Gao et al. No. 53). It results from the release of alkaline ions (in particular Na+ present in excess) by the material that, by combining with atmospheric CO2, gives alkaline species NaHCO3 and Na2CO3 responsible for this phenomenon.
- The increase in the rate of substitution of the starting aluminosilicates by the geopolymer waste acts to increase the concentration of excess sodium ions in the material and consequently the pH.
- Analysis of the results (fig. 12, 13 and 14), microscopic observations (fig. 16) and very low mass losses recorded show the very good resistance of these materials to acid attack.
Response to note 7/ The remark has been taken into consideration and the requested modification is inserted in the text.

Reviewer 2 Report
The paper deals with an important issue, which is the using of waste materials in the process of producing new materials (in accordance with the principles of circular economy).
Despite the fact that the work concerns an important issue, it requires corrections and a broader description of the results obtained:
1. Line 46: replace the dots with other ions.
2. Line 65: decode the abbreviations (FARG and MKRG). First write the full name and the abbreviation in brackets. Describing the abbreviations alone: ​​FA and MK is insufficient.
3. Line 81: write down what it means before the abbreviation (OPC).
4. Line 104: "Lahoti and col" - can replace col for example et al.
5. Line 140: write where the fly ash came from; was it fly ash from coal combustion? Also, add from which power plant it was obtained (if possible).
6. Line 149: Briefly describe in the text the results of the analysis, contained in table 1. Be sure to add in the text that SO3 occurs in combination with CaO (creating eg CaSO4). It is known that SO3 does not occur in powders - it is only the result of the XRF elemental analysis (where we obtain the content of the elements) into oxides by the program.
7. Line 156: write down what it means before L/S. Then this extended description does not need to be used in table 2. The abbreviation alone is sufficient there.
8. Subchapter 2.3 (fire resistance): which equipment was used for the tests for heating to 400, 600, 800 and 1000 degrees? My guess is TG - but it's not clearly written. If it was TG, why is the heating rate 6.67 deg / min (line 176)? Where does this speed come from? Especially that in the description of the TG methodology later in the article, the heating speed is selected as 10 degrees / minute (line 198).
9. Line 193: is RDX. I guess it should be XRD?
10. Line 229: There is TGA / TDA. I guess it should be TGA / DTA? And the captions in figures 2 a, b and c is TDA ... Please correct it.
11. Line 267: "we present". Please replace it with eg "it is presented".
12. Line 316: Please add in figure 6 in caption what temperatures and materials were analyzed.
12. Line 319: Same note as above.
13. Please spread the word about the results presented in Figures 10, 11 and 12.
14. A lot of research has been done - this information cannot be seen in the conclusions. Please expand the conclusions drawn.
15. It would be good to replace pauses in the text with, for example, numbers or letters - it looks better. In general, please improve the visual side of the article. Figure captions, text layout.
16. In the introduction, it is possible to add about the European Union guidelines for the use of waste materials in materials engineering - in accordance with the principle of circular economy.
Author Response
Responses to reviewers’ comments
Review #2
Your remarks n° 2, 3, 4, 5, 6, 7, 9, 10, 11, 12, 13, 14 and 15 have been taken into account. The modifications requested are inserted in the text.
Response to note 1/ The cations mentioned are those of the most used activators for the production of geopolymer materials intended for construction. We have modified the sentence in question.
Response to note 8/ We have clarified the text and corrected the error on the heating rate. The samples were heated at a rate of 6.67°C/min in an electric muffle furnace. Then they were subjected to mechanical tests to determine the compressive strength. For the TGA-DTA tests, the samples were heated at the same rate.
Response to note 16/ Your remark has been taken into account (see the introduction lines 60-64).

Reviewer 3 Report
Review of the manuscript
Influence of the integration of geopolymer wastes on the characteristics of binding matrices subjected to the action of temperature and acid environment
Authors: Rabii Hattaf, Abdelilah Aboulayt, Nouha Lahlou, Mohamed Ouazzani Touhami, Moussa Gomina, Azzedine Samdi, Redouane Moussa
This paper presents the impact of the geopolymer wastes integration up to 50 wt.% on the the characteristics of binding matrices with very interesting mechanical behaviour in acidic environment. The context of the study is well explained with reference to relevant literature and relating research project. This latter aspect is particularly appreciated as it shows the intense research activities around this subject and the multi-disciplinarity of this work. The microstructural characterization tools used in this paper is also appreciated and it is not common to find them in the field of civil engineering. The paper can be considered for publication after the Authors have taken into account these minor comments:
Check English in the whole paper to correct some grammar and language mistakes.
There are also some annoying spelling errors:
Line 28: please remove ‘of’ in ‘containing of geopolymer
Lines 38-39: please reformulate the sentence
Line 76: please reformulate the sentence (enable?)
Line 127: please replace bath by batch
Line 158: please remove ‘on’
Line 196: 0.059 ° instead of 0.059°.
- Line 216: please replace hygroscopic water by free water (the term hygroscopic is not appropriate)
- Lines 84-86: the authors state that ‘These phases dehydrate between 400 and 550 ° C, which cause the formation of pores and the creation of cracks. This deep microstructural damage leads to a drastic degradation of mechanical performance.’ However, concrete is prepared at room temperature! The authors can explain what they mean by this sentences?
- Line 86-88: the authors state that ‘in geopolymers, the water present is physisorbed in nature and is not part of the structure’. This information is not correct because water is also chemisorbed, otherwise, how to explain the hydration phenomenon?
- Line 244: the authors state that ‘No microstructural change is observed after exposure of these materials to 400 ° C,..’. However, we can notice that a crystallization phenomenon occurs (the diffraction peaks become narrow).
This sentence needs to be re-formulated
- In section 2.1: the description of XRF should be grouped with the other techniques in section 2.5
- The description of the XRD refinement procedure is missing
- The images in figure 7 must be improved
- General comment. It would be useful to clarify the number of replicates that have been performed for each test and material composition. This is important as the paper include statistical indications on the obtained properties (e.g. average values, standard deviation, etc.).
Author Response
Responses to reviewers’ comments
Review #3
Your remarks n° 1, 5, 6, 7 and 8 have been taken into account. The modifications requested are inserted in the text.
Response to note 2/ Indeed, ordinary Portland cement hardens at room temperature and geopolymers at temperatures below 100° C. However, in this part of the introduction we compare the fire resistance and the behavior that these materials undergo during a thermal treatment.
Response to note 3/ Regarding your remark, the presence in the DTA curve of a second endothermic effect is associated with the dehydration of zeolite structures which form simultaneously with the geopolymer gel (and not with the dehydration of the gel). Indeed, several authors have reported the formation of very small quantities of zeolite phases (impurities) during the development of metakaolin-based geopolymers (ref no # 59 and 60). See edit lines in text (lines 83-89)
Response to note 4/ The crystallization of refractory phases such as quartz and mullite is not envisaged after the implemented heat treatment. However, the anomaly observed by the reviewer is attributed to the use of different scales. So we made the correction.

Round 2
Reviewer 1 Report
none